Methods

# Quantification of cristae architecture reveals time-dependent characteristics of individual mitochondria

Mayuko Segawa[1,*] , Dane M Wolf[1,2,*], Nan W Hultgren[3], David S Williams[3] , Alexander M van der Bliek[4,5], David B Shackelford[6], Marc Liesa[1,7] , Orian S Shirihai[1,7]

Recent breakthroughs in live-cell imaging have enabled visualization of cristae, making it feasible to investigate the structure–function relationship of cristae in real time. However, quantifying live-cell images of cristae in an unbiased way remains challenging. Here, we present a novel, semi-automated approach to quantify cristae, using the machine-learning Trainable Weka Segmentation tool. Compared with standard techniques, our approach not only avoids the bias associated with manual thresholding but more efficiently segments cristae from Airyscan and structured illumination microscopy images. Using a cardiolipin-deficient cell line, as well as FCCP, we show that our approach is sufficiently sensitive to detect perturbations in cristae density, size, and shape. This approach, moreover, reveals that cristae are not uniformly distributed within the mitochondrion, and sites of mitochondrial fission are localized to areas of decreased cristae density. After a fusion event, individual cristae from the two mitochondria, at the site of fusion, merge into one object with distinct architectural values. Overall, our study shows that machine learning represents a compelling new strategy for quantifying cristae in living cells.

## Introduction

Mitochondria are dynamic, double-membrane–bound organelles (Friedman & Nunnari, 2014; Cogliati et al, 2016). A relatively porous outer mitochondrial membrane encapsulates a protein-dense inner mitochondrial membrane (IMM), which consists of numerous invaginations, called cristae (Palade, 1953). Over the last decade or so, different studies have demonstrated that the molecular machinery of oxidative phosphorylation is concentrated in cristae membranes (Dudkina et al, 2005; Vogel et al, 2006; Strauss et al, 2008; Davies et al, 2011; Wilkens et al, 2013). Nevertheless, directly

probing the functional significance of cristae structure has been hindered by a longtime inability to visualize the IMM in living cells (Jakobs & Wurm, 2014). Although EM can resolve cristae effectively, it is necessary to freeze or fix samples before imaging, which precludes any direct functional readout. Conventional light microscopy, on the other hand, permits live-cell imaging, but the Abbe diffraction limit (~200 nm) has prevented the simultaneous resolution of mitochondrial ultrastructure (Jakobs et al, 2020).

Recent advancements, however, in high- and super-resolution imaging technologies, e.g., Airyscan, structured illumination microscopy (SIM), and stimulated emission depletion (STED) microscopy, have enabled the visualization of cristae in living cells (Huang X et al, 2018; Stephan et al, 2019; Wang C et al, 2019; Wolf et al, 2019; Jakobs et al, 2020; Kondadi et al, 2020; Wolf et al, 2020). Remarkably, cristae appear to exhibit their own dynamic behaviors (Huang X et al, 2018; Stephan et al, 2019; Wang C et al, 2019; Kondadi et al, 2020) and display disparate membrane potentials ($\Delta\Psi_m$), indicating that they function as independent bioenergetic units (Wolf et al, 2019).

Powerful new techniques for visualizing cristae in living cells necessitate proportionately robust methods for quantification. In this study, we present a novel approach using the open-source plugin, Trainable Weka Segmentation (TWS) (Hall, 2009; Arganda-Carreras et al, 2017), to measure cristae. This semi-automated method is beneficial because it avoids the bias and inefficiency associated with manual segmentation (Caffrey et al, 2019) and is able to detect significant differences in cristae density and architecture, resulting from pathological changes in IMM integrity. Our approach is also sufficiently sensitive to measure the remodeling of individual cristae within the same mitochondrion as well as shed light on the dynamic changes in the IMM during mitochondrial fusion and fission.

Numerous studies have demonstrated a link between perturbed cristae structure and a variety of human diseases and medical complications–ranging from diabetes to liver steatosis, from ischemia

[1]Department of Medicine, and Department of Molecular and Medical Pharmacology, David Geffen School of Medicine, University of California Los Angeles, Los Angeles, CA, USA [2]Graduate Program in Nutrition and Metabolism, Graduate Medical Sciences, Boston University School of Medicine, Boston, MA, USA [3]Departments of Ophthalmology and Neurobiology, Stein Eye Institute, David Geffen School of Medicine, University of California, Los Angeles, Los Angeles, CA, USA [4]Molecular Biology Institute at University of California, Los Angeles, Los Angeles, CA, USA [5]Department of Biological Chemistry, David Geffen School of Medicine at University of California, Los Angeles, Los Angeles, CA, USA [6]Department of Pulmonary and Critical Care Medicine, David Geffen School of Medicine, University of California, Los Angeles, Los Angeles, CA, USA; Jonsson Comprehensive Cancer Center, David Geffen School of Medicine, University of California, Los Angeles, Los Angeles, CA, USA [7]Department of Molecular and Medical Pharmacology, David Geffen School of Medicine, University of California, Los Angeles, Los Angeles, CA, USA

Correspondence: mliesa@mednet.ucla.edu; OShirihai@mednet.ucla.edu
*Mayuko Segawa and Dane M Wolf contributed equally to this work

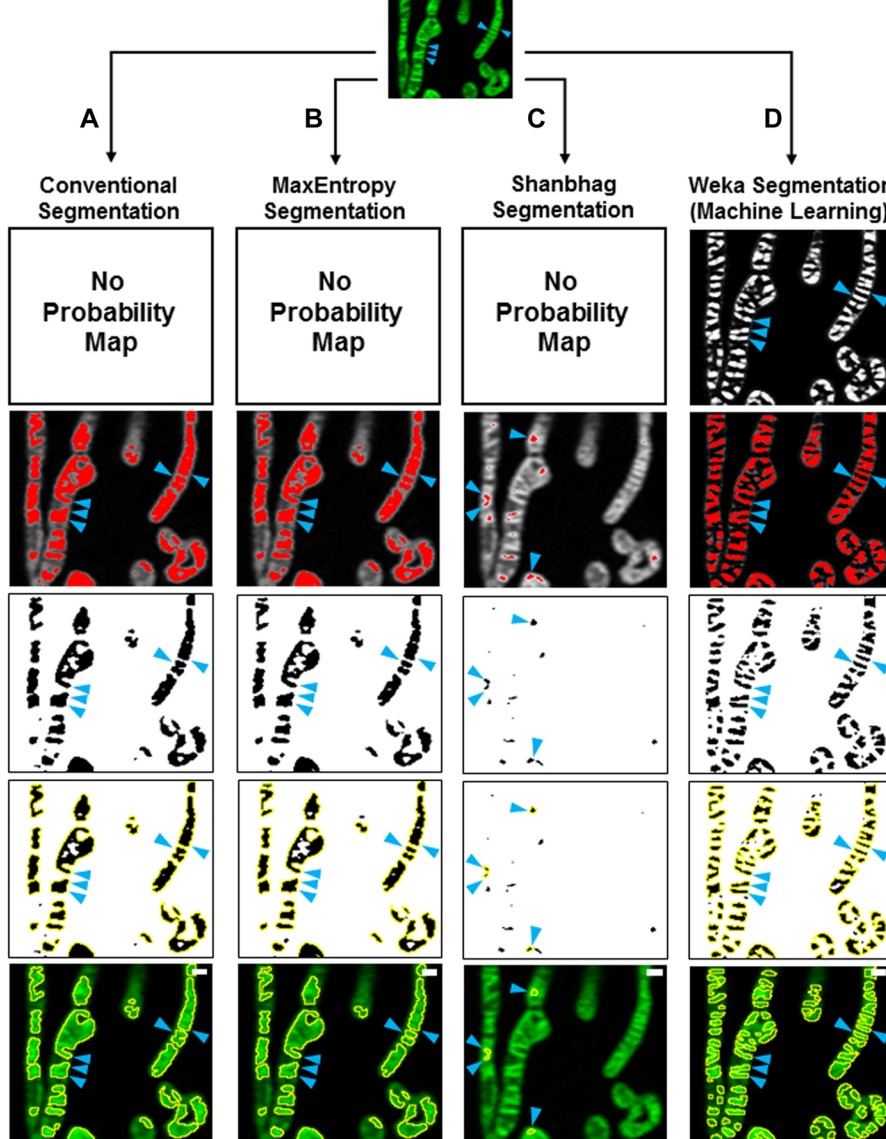

**Figure 1. Trainable Weka Segmentation protocol more effectively segments mitochondrial cristae from living cells compared with various thresholding standards.**
**(A, B, C, D)** Step-wise comparison of different segmentation workflows: Conventional (manual) (A); MaxEntropy (B); Shanbhag (C); and Trainable Weka Segmentation (D). Note that without a probability map, Conventional, MaxEntropy, and Shanbhag thresholding, which depend exclusively on pixel intensities, cannot effectively segment cristae. **(A, B, C, D)** Blue arrowheads denote successful segmentation of cristae (column D) versus ineffective segmentation (columns A, B, C). Scale bars = 500 nm.

reperfusion injury to even aging (Acehan et al, 2007; Amati-Bonneau et al, 2008; Zick et al, 2009; Sivitz & Yorek, 2010; Birk et al, 2013; Daum et al, 2013; Mamikutty et al, 2015; Vincent et al, 2016). Our TWS approach provides a novel platform for directly probing the role of IMM architecture in normal as well as dysfunctional mitochondria and represents a new tool for deciphering the complex relationship between mitochondrial membranes and organismal homeostasis.

## Results

### TWS is more effective at quantifying cristae in living cells compared with conventional thresholding techniques

To quantify cristae in a high-throughput, semi-automated way, we leveraged the open-source machine-learning TWS plugin, available

in Fiji (for more details, see specific steps in the Materials and Methods section). We first addressed whether our machine-learning protocol was more effective than traditional thresholding at quantifying cristae in living cells. Using the same images from multiple experiments, we compared our novel TWS method with conventional (i.e., manual) thresholding or different thresholding algorithms, which rely on separating high-intensity pixels from low-intensity pixels as a way to distinguish regions of interest (ROIs) from background. Compared with images analyzed with our TWS protocol (Fig 1D), we found that conventional thresholding was unable to differentiate between cristae less than ~200 nm apart from one another, leading to large ROIs appearing to contain multiple structures (Fig 1A). We also compared our TWS protocol with more restrictive thresholding algorithms (e.g., MaxEntropy and Shanbhag), available in Fiji (Fig 1B and C), and we found that our machine-learning approach was more effective than even restrictive

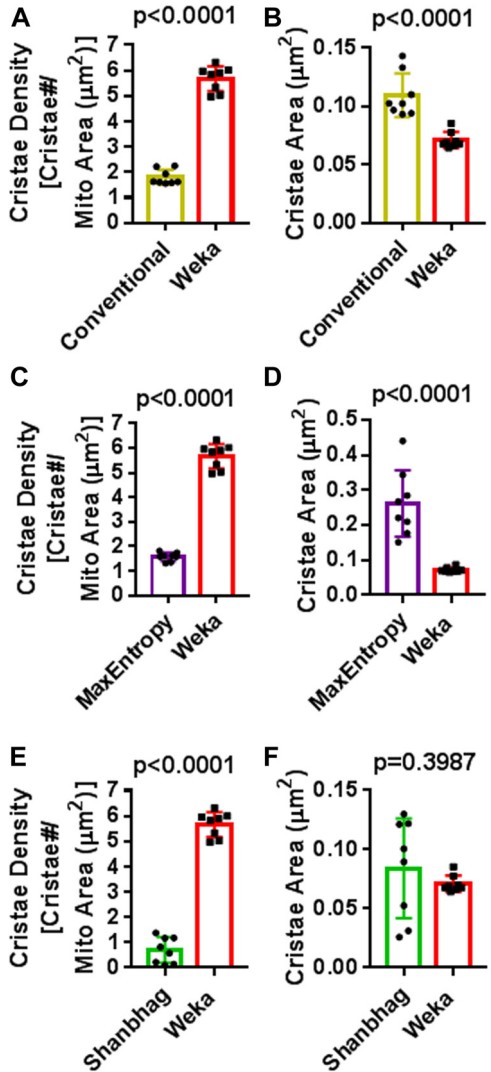

**Figure 2. Trainable Weka Segmentation (TWS) protocol offers significant advantages over standard thresholding techniques for the quantification of cristae in living cells.**

Quantification of cristae parameters using TWS protocol versus standard thresholding. **(A, C, E)** Quantification of cristae density (cristae#/$\mu m^2$) in HeLa cells stained with NAO. Note: TWS is significantly more effective at measuring cristae density compared with Conventional, MaxEntropy, and Shanbhag segmentation methods, respectively. N = 8 independent experiments. **(B, D, F)** Quantification of cristae area ($\mu m^2$) in HeLa cells stained with NAO. Note: Conventional and MaxEntropy segmentation are unable to segment cristae effectively, resulting in significantly higher values in cross-sectional area, compared with Weka segmentation. Conversely, Shanbhag segmentation shows average cristae areas similar to those of Weka segmentation, but this results from Shanbhag being overly restrictive, subsequently underestimating or entirely missing a large proportion of cristae structures (see Fig 1). N = 8 independent experiments. Data information: Data are presented as mean ± SD. *P*-values are shown in panels (*t* tests).

algorithms at segmenting cristae. Using our TWS method to first generate a probability map, we were able to detect significantly more cristae per mitochondrial area (Fig 2A, C, and E) as well as significantly smaller cristae (Fig 2B and D). We observed that Shanbhag thresholding had cristae areas similar to those measured by our TWS protocol (Fig 2F). Inspection of the images indicated that this was due

to the overly restrictive properties of the Shanbhag algorithm, which tended to underestimate the actual cristae areas.

## TWS protocol for segmenting cristae is applicable to multiple cell types and is compatible with different mitochondrial dyes

After determining that our machine-learning method for quantifying cristae was more effective than conventional thresholding techniques, we addressed whether we could effectively segment the ultrastructure in a variety of cell types. Performing live-cell Airyscan imaging of mitochondrial ultrastructure in HeLa, L6, H1975, and HUH7 cells, we found that our TWS protocol appeared to be effective at segmenting cristae, regardless of the cell type (Fig 3A–D, respectively). Intriguingly, we observed that, although there appeared to be no differences in cristae density (Fig 3E), cristae in HeLa cells tended to have a smaller cross-sectional area (Fig 3F) as well as a larger aspect ratio (Fig 3G), compared with the cristae in the other cell types. Next, we determined that it was not only feasible to segment cristae from mitochondria stained with NAO but also with other mitochondrial dyes, such as Rho123 (Fig S1A) and MTG (Fig S1B).

## TWS protocol can quantify differences in cristae density, area, and shape in cell-culture model of IMM perturbation

Next, we tested whether our TWS protocol could be used to discern defects in mitochondrial ultrastructure associated with pathological changes to mitochondrial membrane integrity. To examine this question, we used H1975 cells deficient in the phosphatase PTPMT1 (Wolf et al, 2019), which converts phosphatidylglycerolphosphate into phosphatidylglycerol, a critical precursor of the signature mitochondrial phospholipid, cardiolipin (Zhang et al, 2011). Previous studies have shown by EM that loss of PTPMT1 leads to severe derangement of the IMM, where cristae appear swollen and disorganized (Zhang et al, 2011). Compared with control H1975 cells (Fig 4A and C), our live-cell imaging of PTPMT1-deficient H1975 cells showed mitochondria with defective ultrastructure (Fig 4B and D). Remarkably, our machine-learning method was able to quantify these perturbations, revealing a significant decrease in cristae density (Fig 4E), increased cristae area (Fig 4F), and decreased cristae aspect ratio (Fig 4G).

## TWS protocol is effective at segmenting cristae in live-cell SIM images, highlighting heterogeneity of mitochondrial ultrastructure within the same organelle

Given the increasing availability of high- and super-resolution imaging technologies, we tested whether our TWS protocol could be advantageous for quantifying cristae using other imaging approaches than Airyscan. Staining HeLa cells with MTG, we performed live-cell SIM imaging (Fig 5A), and we observed mitochondrial ultrastructure in even finer detail than with Airyscan (Fig 5B). We then compared our TWS protocol with a conventional thresholding approach, and determined that, as with Airyscan imaging, machine learning provided significant advantages for quantifying cristae in living cells (Fig S2A–D). We observed that our TWS protocol could segment a variety of cristae structures. Remarkably, cristae within

**Life Science Alliance**

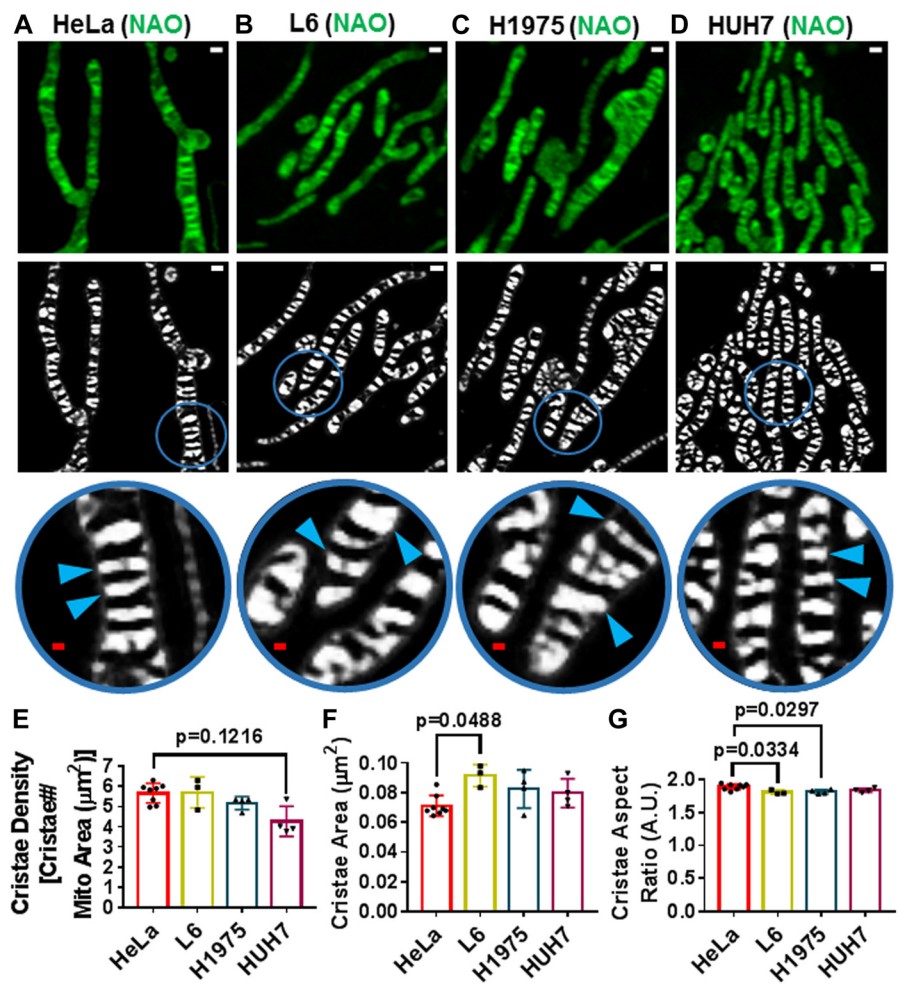

**Figure 3. Trainable Weka Segmentation protocol enables segmentation of cristae in a variety of cell types.**
**(A, B, C, D)** Live-cell Airyscan images of HeLa (A), L6 (B), H1975 (C), and HUH7 (D) mitochondria, stained with NAO. Note that top row shows original images (scale bars = 500 nm). Bottom row, including circular, zoomed-in regions, show probability maps of cristae in respective cell types (red scale bars = 200 nm). Blue arrowheads denote cristae. **(E)** Quantification of cristae density between HeLa, L6, H1975, and HUH7 cells, stained with NAO. N ≥ 3 independent experiments. **(F)** Quantification of cristae area between HeLa, L6, H1975, and HUH7 cells, stained with NAO. N ≥ 3 independent experiments. **(G)** Quantification of cristae aspect ratio between HeLa, L6, H1975, and HUH7 cells, stained with NAO. N ≥ 3 independent experiments. Note that HeLa cells tend to have lower cross-sectional area of cristae together with increased aspect ratio. Data information: Data are presented as mean ± SD. *P*-values are shown in panels (ANOVA).

the same mitochondrion often exhibited heterogeneous morphologies, ranging from classical, lamellar structures to arches, and even to apparently interconnected jigsaw configurations (Videos 1 and 2). Moreover, a region of a mitochondrion, appearing to have recently undergone a fusion event, exhibited cristae spanning the interface between the previously separate organelles (Fig 5C).

Because mitochondria are dynamic, frequently changing their morphologies, we next sought to determine if our TWS protocol could segment cristae from organelles with markedly different shapes and sizes. Generally, we observed that our machine-learning approach could segment cristae regardless of the gross mitochondrial architecture: for example, we segmented cristae from thin or more distended organelles (Fig 6A), ouroboros mitochondria (i.e., organelles exhibiting head-to-tail fusion) (Fig 6B), elongated mitochondria with a terminal ouroboros-like structure encompassing a punctate mitochondrion (Fig 6C), fragmented mitochondria (Fig 6D), and mitochondria of intermediate length (Fig 6E). Notably, within these diverse mitochondrial forms, we continued to observe heterogeneity in cristae structure within the same organelle.

Having demonstrated our ability to effectively segment a range of cristae shapes from diverse mitochondrial morphologies, we next

used our TWS protocol to estimate the average number of cristae in a typical HeLa cell. Analyzing z-stacks of SIM images, we determined that there were 187.2 ± 91 mitochondria and 1,415.6 ± 257 cristae per HeLa cell. The average number of cristae per mitochondrion, therefore, was 9.7 ± 4. Given that the lateral resolution of SIM is ~100 nm, it should be noted that cristae positioned less than 100 nm apart would not be resolved. Thus, these measurements likely underestimate the actual values of cristae per cell as well as cristae per mitochondrion. Using live-cell imaging technologies with higher resolution, in conjunction with our TWS protocol, would probably improve the accuracy of these measurements.

## TWS protocol can quantify acute changes in mitochondrial ultrastructure, resulting from FCCP treatment

We next determined whether our machine-learning method could be used for measuring changes in mitochondrial ultrastructure, stemming from acute pharmacological perturbation. Treating HeLa cells with trifluoromethoxy carbonylcyanide phenylhydrazone (FCCP), we imaged them for ~1 h. Consistent with

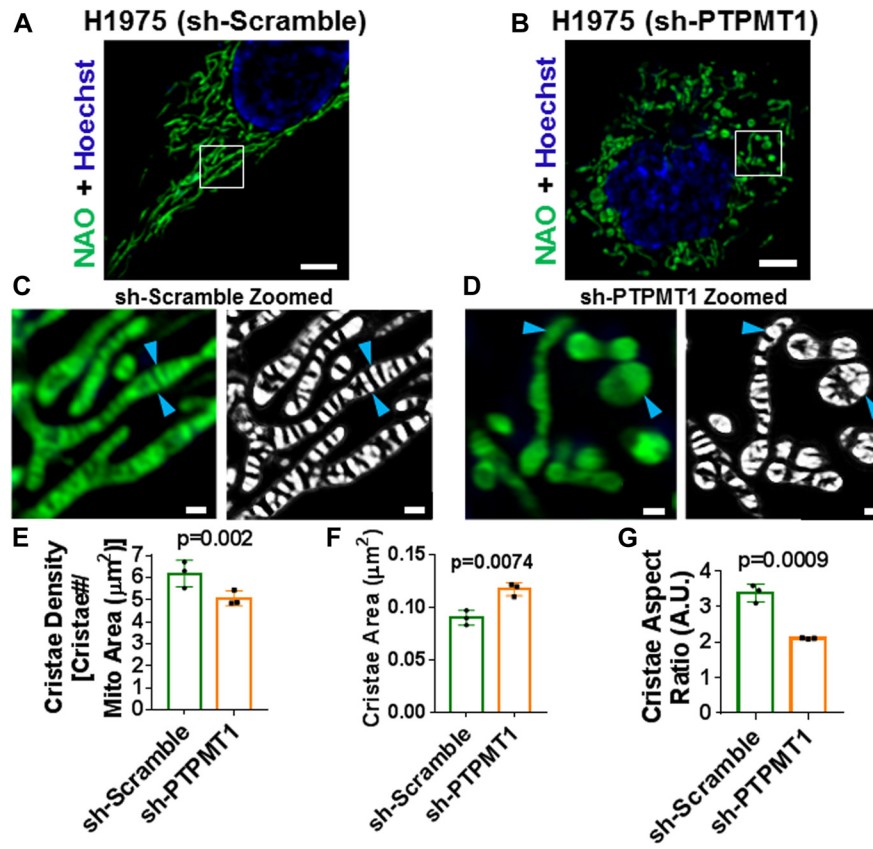

**Figure 4. Trainable Weka Segmentation protocol is sufficiently sensitive to detect differences in cristae density, area, and shape in cardiolipin-deficient (sh-PTPMT1) H1975 cells, a model of inner mitochondrial membrane dysregulation.**
**(A)** Image of sh-Scramble H1975 cell, stained with NAO and Hoechst. Scale bar = 5 μm. **(B)** Image of sh-PTPMT1 H1975 cell, stained with NAO and Hoechst. Scale bar = 5 μm. **(C)** Zoomed-in region of (A), showing normal lamellar cristae (blue arrowheads) in original image (left) compared with cristae probability map (right). Scale bars = 500 nm. **(D)** Zoomed-in region of (B), showing deranged cristae (blue arrowheads) in original image (left) compared with cristae probability map (right). Scale bars = 500 nm. Note that these images appear to corroborate previously published EM data, showing loss of PTPMT1 results in swelling and disruption of cristae structure. **(E)** Quantification of cristae density between sh-Scramble and sh-PTPMT1 in H1975 cells, stained with NAO. N = 3 independent experiments. **(F)** Quantification of cristae area between sh-Scramble and sh-PTPMT1 in H1975 cells, stained with NAO. N = 3 independent experiments. **(G)** Quantification of cristae aspect ratio between sh-Scramble and sh-PTPMT1 in H1975 cells, stained with NAO. N = 3 independent experiments. Data information: Data are presented as mean ± SD. *P*-values are shown in panels (*t* tests).

previous studies, we observed time-dependent changes in both gross as well as fine structure (Fig 7A). FCCP treatment tended to significantly decrease cristae density (Fig 7B) and cristae area (Fig 7C) and significantly increased cristae circularity, resulting from an apparent rise in the number of circular cristae membranes (Fig 7D). Interestingly, we observed that FCCP treatment did not, on average, decrease the cristae aspect ratio (Fig S3B), despite the relatively larger number of circular cristae. This appeared to be due to the concomitant formation of cristae with unusually high aspect ratios, resulting from extreme mitochondrial swelling, associated with the collapse of the proton gradient (Fig S3A).

### TWS protocol can measure real-time changes in cristae density, size, and shape

We next performed time-lapse SIM imaging of mitochondria in HeLa cells stained with MTG to determine whether we could track changes in mitochondrial ultrastructure at high temporal as well as spatial resolution. Imaging mitochondria at ~2-s intervals, we observed instances of parallel lamellar cristae fusing into arched structures (Fig 8A and Video 3), and we measured around twofold changes in aspect ratio. We also detected events where a series of lamellar cristae appeared to undergo fission and rapidly remodel into arching structures, running parallel to the long axis of the organelle (Fig 8B and Video 4). These alterations in shape typically reflected a two to threefold change in circularity. Ultrastructural remodeling frequently

appeared to be associated with fusion and fission events, occurring within relatively short time intervals (i.e., approximately every few seconds), as reflected by dynamic changes in cristae circularity (Fig 8C and Video 5).

Given the relatively rapid remodeling of mitochondrial ultrastructure that we observed, we wanted to estimate the dynamic ranges of these parameters in individual mitochondria over a brief period of time. We, therefore, used our TWS protocol to quantify the cristae density, area, circularity, aspect ratio, and total number per mitochondrion at ~2-s intervals for roughly 30 s (Fig 9A–E). Fig 9F shows the mitochondrion corresponding to the green curves in Fig 9A–E. In general, we observed that cristae density, size, and shape changed over time within relatively narrow ranges between different organelles within the same cell. We then expressed these time-dependent alterations as SDs per min to summarize their dynamic ranges (Fig 9G).

### Real-time quantification of cristae remodeling during mitochondrial fusion and fission highlights the dynamic nature of mitochondrial membranes

Finally, we examined whether our TWS protocol could quantify cristae during classical examples of mitochondrial dynamics. During mitochondrial fission events, we observed that the site of membrane constriction and fission appeared to consist of inner boundary membranes rather than cristae membranes (Fig 10A–C

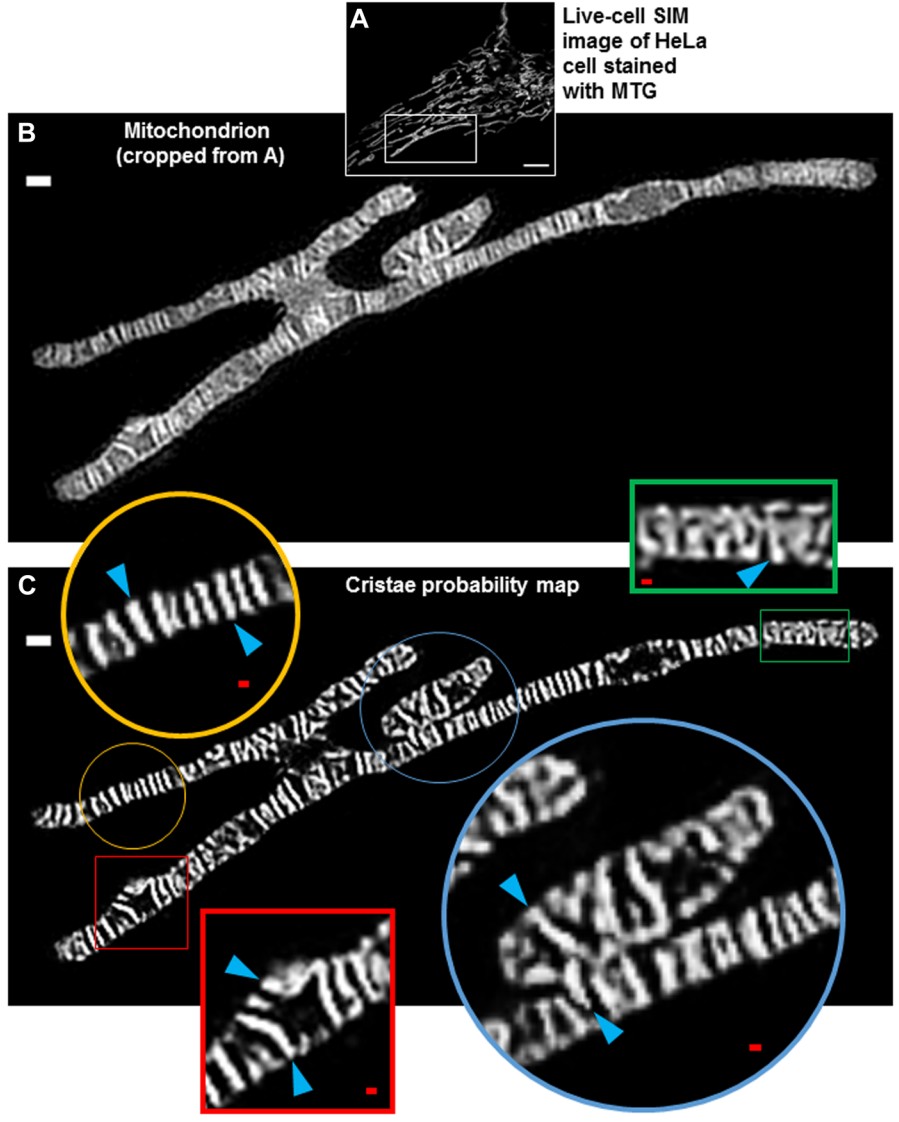

**Figure 5. Trainable Weka Segmentation is effective at segmenting cristae in live-cell images obtained with structured illumination microscopy (SIM), highlighting ultrastructural heterogeneity within the same mitochondrion.**
**(A)** Live-cell SIM image of HeLa cell stained with MTG. Scale bar = 5 $\mu$m. **(B)** Mitochondrion cropped from (A), showing fine structure of mitochondrion. Scale bar = 500 nm. Note that the different mitochondrial regions appear to encompass a single fused structure. **(C)** Cristae probability map of cropped SIM image from (B). White scale bar = 500 nm. Note that the zoomed-in regions show heterogeneous cristae architecture (blue arrowheads) within the same organelle: the gold circle highlights a region of lamellar cristae; the red square shows a variety of arched cristae, running either parallel or perpendicular to the long axis of the organelle; the green rectangle shows a jigsaw configuration; and the blue circle shows cristae spanning adjacent mitochondrial structures. Red scale bars = 100 nm.

and Videos 6–8). Because our TWS protocol was trained to recognize cristae, rather than inner boundary membrane, it is particularly apparent in the TWS cristae probability maps that cristae are not localized to the fission sites.

We next applied our TWS protocol to mitochondria engaging in fusion events. Intriguingly, immediately before membrane fusion of the two organelles, we observed finger-like protrusions from the ends of one of the mitochondria (Figs 11A, S4A, and Videos 9 and 10), which appeared to bridge the membranes of the two organelles. This observation supports recently published models of Mgm1/Opa1–mediating IMM fusion through the formation of highly curved membrane tips (Yan et al, 2020). Using live-cell SIM imaging, we observed the formation of such IMM protrusions on one, rather than both, of the fusing mitochondria. This mode of mitochondrial fusion is compatible with a kind of heterotypic membrane fusion, where the minimum requirement is Opa1 on one side and cardiolipin on the other side of the fusing membranes (Ban et al, 2017). Seconds after the membranes from the two mitochondria fused, we observed fusion of previously separate cristae, originating from the different mitochondria (Fig 11B). After cristae fusion, we measured a more than twofold decrease in cristae circularity, marking a transition to a more branching architecture.

Altogether, our data show that machine learning provides an effective way to quantify cristae density, size, and shape in living cells, representing a powerful new tool for investigating cristae structure and function in real time.

## Discussion

Since the discovery of the mitochondrial cristae in the mid-20[th] century (Palade, 1953), imaging these structures in real time has remained a formidable challenge. Because of the necessity of freezing or fixing samples before imaging, EM is incompatible with probing cristae in living cells. Conversely, whereas conventional optical microscopy has permitted live-cell imaging of mitochondria,

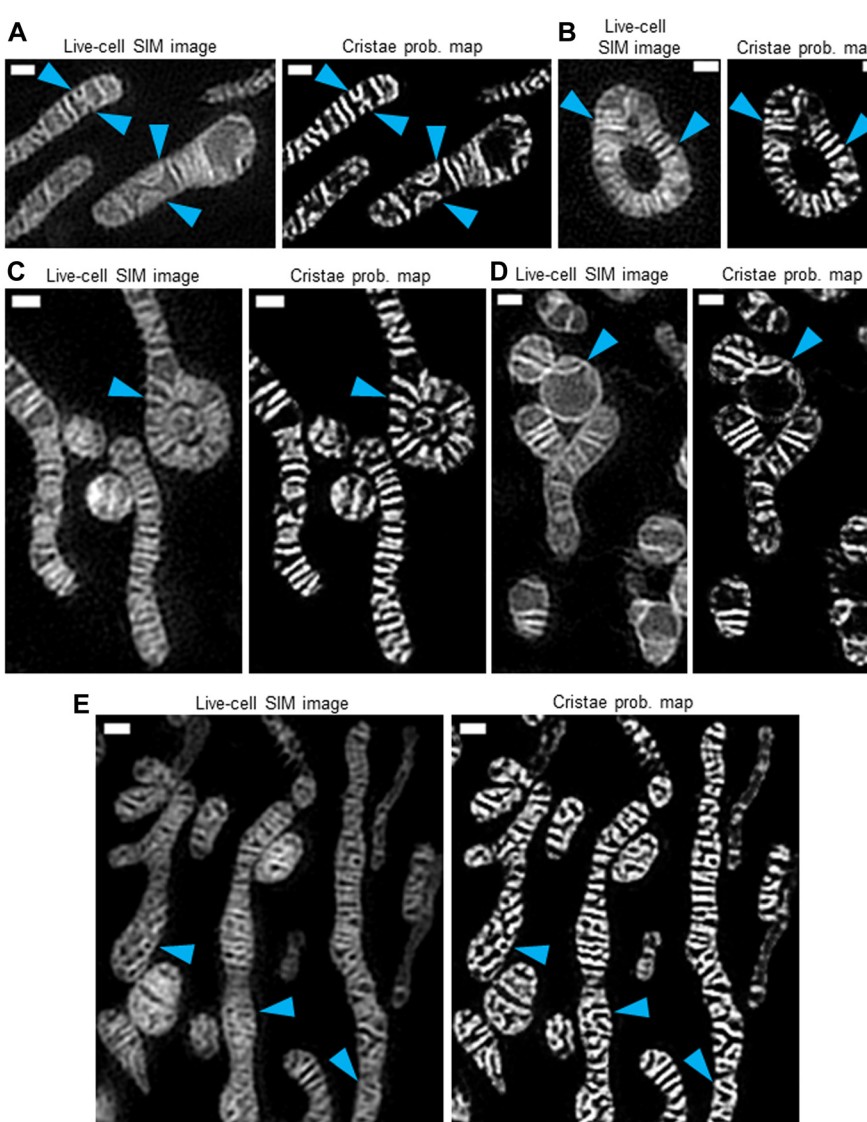

**Figure 6. Trainable Weka Segmentation protocol can segment different cristae structures found in a range of mitochondrial morphologies.**
Live-cell SIM images of HeLa cells stained with MTG (left) and cristae probability maps (right) showing various mitochondria with differing cristae structures. **(A)** Thin and distended mitochondria, showing relatively small and large arched cristae, respectively (blue arrowheads). Scale bars = 500 nm. N = 3 independent experiments. **(B)** Ouroboros mitochondrion, showing cristae radiating from central gap (blue arrowheads). Scale bars = 500 nm. N = 3 independent experiments. **(C)** Elongated mitochondrion with ouroboros-like end, containing cristae radiating from center (blue arrowhead); note the spherical mitochondrion filling the gap within this ouroboros-like structure. Scale bars = 500 nm. N = 3 independent experiments. **(D)** Fragmented mitochondria, showing arched cristae structures (blue arrowhead). Scale bars = 500 nm. N = 3 independent experiments. **(E)** Mitochondria of intermediate length, containing various netlike and/or curving cristae (blue arrowheads). Scale bars = 500 nm. N = 3 independent experiments.

the Abbe diffraction limit has wholly obscured the intricacies of the IMM. Recent advancements in high- and super-resolution imaging technologies, however, have enabled the visualization of cristae in living cells, leading to an appreciation that cristae are not static structures but appear to possess their own dynamics comparable with the fusion and fission that regulate the plasticity of the larger mitochondrial network (Huang X et al, 2018; Stephan et al, 2019; Giacomello et al, 2020; Kondadi et al, 2020). Furthermore, we recently showed that cristae are functionally independent bioenergetic compartments, capable of preventing the spread of localized damage (Wolf et al, 2019).

The emerging interest in imaging cristae in living cells calls for novel segmentation methods that will promote accurate and efficient quantification. To address this growing demand, we used the open-source machine-learning plugin, TWS, to develop a new approach to segment cristae. Here, we showed that this method is not only more effective than common thresholding techniques available in Fiji but also sufficiently robust to pick out differences in cristae parameters in various cell lines. Furthermore, we demonstrated that, using our method, it is possible to quantify pathological changes in mitochondrial ultrastructure, resulting from genetic or pharmacological perturbations. Using various filtering algorithms, instead of pixel intensities alone, to segment cristae, results in significantly more accurate measurements; moreover, the semi-automation of our TWS protocol makes it feasible to efficiently analyze large numbers of cristae (e.g., 1,000 or more per cell). We also showed that our TWS protocol can be particularly valuable for probing the dynamic nature of the IMM. Our quantifications of real-time changes in cristae density, size, and shape are consistent with recent observations of cristae remodeling in living cells (Huang X et al, 2018; Stephan et al, 2019; Wang C et al, 2019; Kondadi et al, 2020). Furthermore, the remarkable heterogeneity in cristae structures that we observed in this study provides further mechanistic insight into the heterogeneity in membrane potential among different cristae within the same mitochondrion (Wolf et al, 2019).

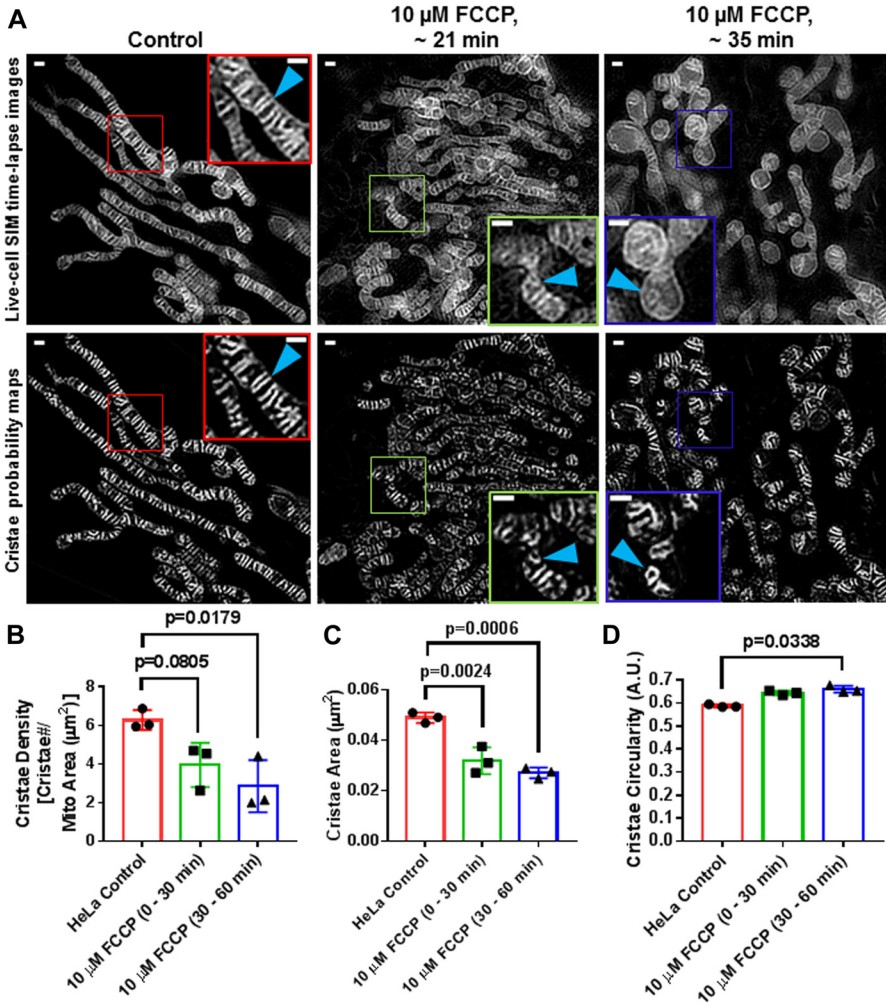

**Figure 7. Trainable Weka Segmentation protocol can capture changes in cristae density, area, and shape after acute treatment with FCCP.**
**(A)** Live-cell SIM images of HeLa cells stained with MTG (top row) together with cristae probability maps (bottom row). Note that the left-most column shows a representative image of control, whereas the center and right-most columns show representative images of cells treated with 10 $\mu$M FCCP between 0 and 30 min and 30 and 60 min, respectively. Scale bars = 500 nm. N = 3 independent experiments. **(B, C, D)** Quantification of SIM images from (A), showing a time-dependent decrease in cristae density (B) and cristae area (C), as well as an increase in cristae circularity (D) as a result of the FCCP treatment. N = 3 independent experiments. Data information: Data are presented as mean ± SD. *P*-values are shown in panels (ANOVA).

It is important to note that, although state-of-the-art high- and super-resolution live-cell imaging technologies are providing exciting new windows onto the complex biology of mitochondrial membranes, they remain, to date, unable to supersede EM as a way to map the spatial dimensions of the organelle. For example, the resolution of electron tomography (ET) systems typically ranges from 5 to 20 nm. Therefore, given average ET measurements of crista length and width to be 240 and 20 nm, respectively, the cross-sectional area of a crista is 4,800 nm$^2$ (or 0.0048 $\mu$m$^2$) (Mannella et al, 2013). The lateral resolution of SIM (including the GE DeltaVision OMX SR system used in this study) is ~100 nm. Our measurements of mean crista area taken from SIM micrographs were 0.0489 $\mu$m$^2$. This value is about one order of magnitude larger than estimates obtained from ET micrographs, a disparity consistent with an ~10-fold difference in the resolution of the two imaging technologies. Therefore, researchers should bear in mind that measurements of cristae area from live-cell, super-resolution images are likely to be overestimated by roughly a factor of 10. Given that the mean crista-to-crista distance in HeLa cells, as measured by EM, is roughly 51–120 nm (Wilkens et al, 2013; Stephan et al, 2019), it is likely that our Airyscan and SIM images did not fully capture all of the cristae within a particular frame–especially in cases where cristae were positioned more closely together than could be resolved by the imaging

systems used in this study. Using diffraction-unlimited nanoscopy (e.g., STED) to image cristae in living cells will likely decrease, if not entirely remove, such disparities. Incidentally, the problem of fully segmenting cristae within a mitochondrion is comparable with the difficulty of completely segmenting mitochondria within a cell, which remains a significant challenge, despite the larger dimensions of the organelles.

Although our TWS protocol for quantifying cristae is necessarily constrained by the resolving power of the microscopes used to image the IMM, it remains particularly useful for obtaining relative measures of the effects of cristae perturbation, as we demonstrated in our genetic and pharmacological models. The actual dimensions of the cristae parameters may be over- or underestimated, according to the specific measurement in question, but tracking relative changes can nevertheless yield valuable information about the effects of potential modulators of cristae density, size, and shape.

Overall, in this study, we present a novel approach to further illuminate the dynamic nature of mitochondrial membranes by quantifying changes in cristae density and architecture in real time. Future studies examining the relationship between cristae structure and function will likely benefit from leveraging a machine-learning segmentation protocol like the one we outlined here.

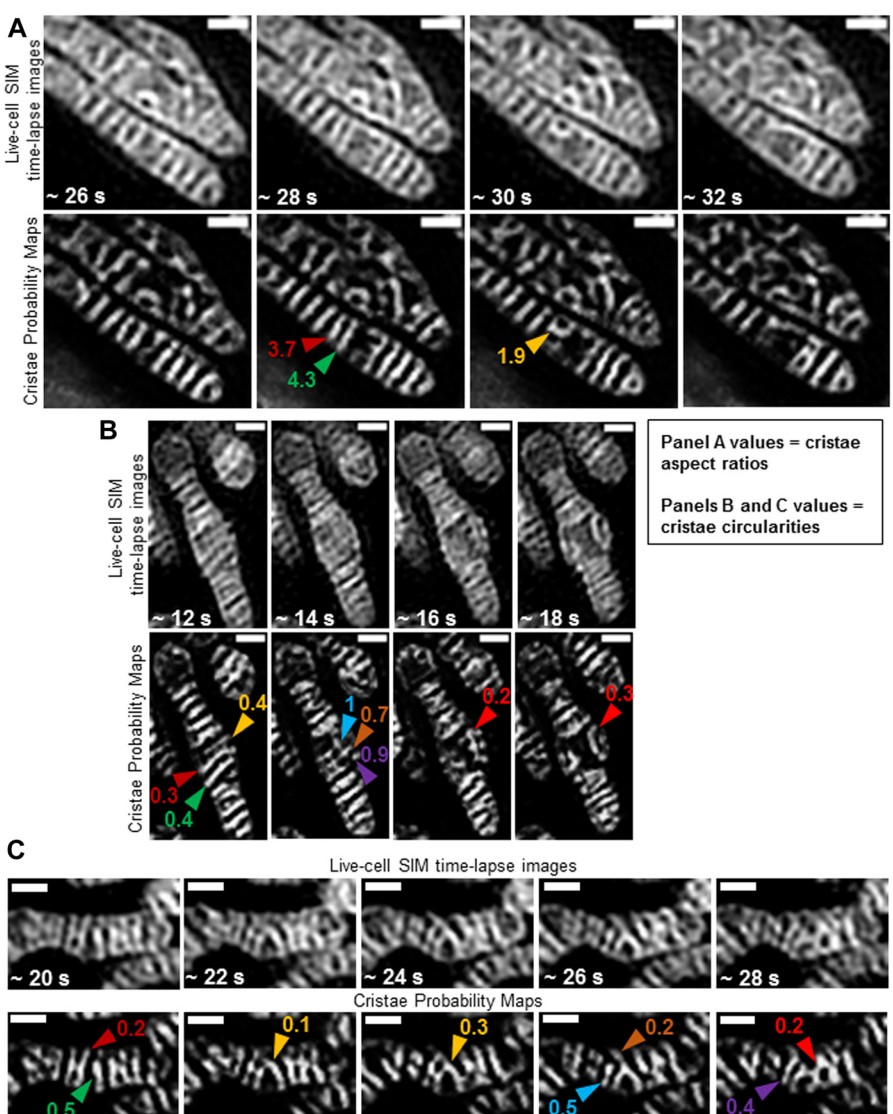

**Figure 8. Trainable Weka Segmentation protocol can quantify real-time cristae remodeling within the same mitochondrion.**
Zoomed-in, time-lapse SIM images of HeLa cells stained with MTG (top rows) and cristae probability maps (bottom rows). **(A)** Example of quantification of shape changes before and after cristae fusion event. Note parallel cristae (red and green arrowheads) at ~28 s are separate structures with aspect ratios of 3.7 and 4.3, respectively; but, after fusing into an arched structure (gold arrowhead), the aspect ratio of the resulting crista is altered to 1.9. Scale bars = 500 nm. **(B)** Example of quantification of shape change before and after cristae fission events. Note 3 parallel cristae (gold, red, and green arrowheads) at ~12 s have a circularity of 0.4, 0.3, and 0.4, respectively. After fission of these cristae at ~14 s, however, membrane fragments (blue, brown, and purple arrowheads) have a circularity of 1, 0.7, and 0.9, respectively. At ~16 and 18 s, these fragments appear to fuse into a branched and arching structure (red arrowheads), respectively, with a circularity of 0.2 and 0.3. Scale bars = 500 nm. **(C)** Example of quantification of multiple, consecutive cristae remodeling events. Note that separate cristae (red and green arrowheads) at ~20 s have a circularity of 0.2 and 0.5, respectively. At ~22–24 s, these cristae appear to fuse (gold arrowheads), showing alterations in circularity to 0.1 and 0.3, at respective time points. This structure, at ~26 s, then appears to divide into a smaller crista (blue arrowhead) with a circularity of 0.5 and a larger forked crista (brown arrowhead) with a circularity of 0.2. At ~28 s, the smaller crista appears to have fused with the upper region of the previously forked crista, resulting in a new structure (purple arrowhead) with a circularity of 0.4, whereas the lower region of the previously forked crista appears to have fused with the crista to the right, generating a more complex network (red arrowhead) with a circularity of 0.2. Scale bars = 500 nm.

## Materials and Methods

### Cell culture

sh-Scramble and sh-PTPMT1 H1975 cells were cultured in RPMI-1640 (31800-022), supplemented with sodium bicarbonate, penicillin/ streptomycin, sodium pyruvate, Hepes, and 10% FBS and grown in 5% $CO_2$ at 37°C. L6, HUH7, and HeLa cells were grown in DMEM (12100-046) and supplemented with sodium bicarbonate, penicillin/streptomycin, sodium pyruvate, Hepes, and 10% FBS and cultured in 5% $CO_2$. KD of PTPMT1 was performed as we previously described (Wolf et al, 2019).

### Live-cell imaging

#### Airyscan
CELLview four-compartment glass-bottom tissue culture dishes (627870; PS, 35/10 mm; Greiner Bio-One) were used for imaging cells.

100 nM 10-*N*-nonyl acridine orange (NAO), 5 μM Rho123, or 200 nM MitoTracker Green (MTG) (Invitrogen) were added to cell culture media and incubated 1–3 h before live-cell imaging. The α Plan-Apochromat 100×/1.46 Oil DIC M27 objective on the Zeiss LSM 880 with Airyscan was used for imaging. Before image analysis, raw .czi files were automatically processed into deconvolved Airyscan images using the Zen software.

#### SIM
The GE DeltaVision OMX SR system with a 60× oil-immersion lens was used to conduct structured illumination super-resolution microscopy. Before live-cell imaging, the system was equilibrated to 37°C with 10% $CO_2$ in humidified chamber. Samples were imaged in regular growth media. Immersion oil with refractive index of 1.522 was used. MTG was excited with the 488-nm laser. For z-stacks, section thickness was set to the 0.125 μm (optimal). For time-lapse imaging, a single plane was imaged at ~2-s intervals.

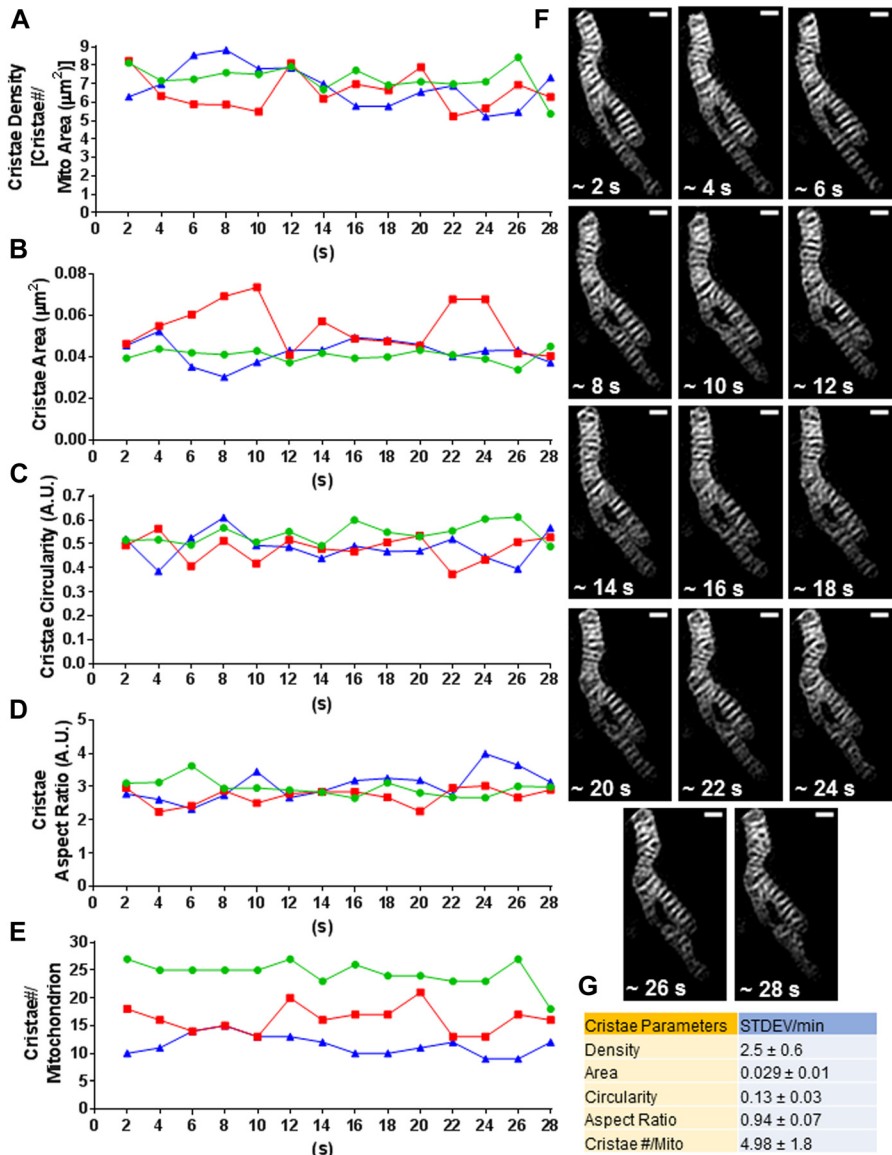

**Figure 9. Quantification of dynamic ranges of cristae parameters within individual mitochondria inside the same HeLa cell.**
**(A, B, C, D, E)** Measurement of time-dependent changes in cristae density (A), cristae area (B), cristae circularity (C), cristae aspect ratio (D), and cristae number per mitochondrion (E). Note that each colored line (red, green, and blue) represents time-dependent changes in cristae parameters within a whole and separate mitochondrion inside the same cell. **(F)** Representative mitochondrion from (A, B, C, D, E). Scale bars = 500 nm. Note that the different values associated with this mitochondrion are displayed by the green curves. **(G)** Table showing SDs in cristae density, area, circularity, aspect ratio, and cristae number per mitochondrion per min. Note that the SDs in these cristae parameters reflect typical, time-dependent changes in cristae density and architecture. N = 3 independent experiments. Values are shown with associated SDs.

| Cristae Parameters | STDEV/min |
|---|---|
| Density | 2.5 ± 0.6 |
| Area | 0.029 ± 0.01 |
| Circularity | 0.13 ± 0.03 |
| Aspect Ratio | 0.94 ± 0.07 |
| Cristae #/Mito | 4.98 ± 1.8 |

## Image analysis

Processed Airyscan as well as SIM images were analyzed with ImageJ (Fiji) software, National Institutes of Health. Before cell cropping and quantification, background was subtracted from images using a rolling ball filter = 50. After developing analysis procedures, we built macros for high-throughput image quantification. For representative images in figures, we adjusted pixel intensities to optimally show relevant changes in IMM structure.

## Statistical analysis

Statistical analysis was performed on GraphPad Prism and Microsoft Excel. Independent experimental data sets were subjected to D'Agostino–Pearson omnibus and/or Shapiro–Wilk normality tests to assess whether data were normally distributed.

Data were subjected to parametric or nonparametric two-tailed $t$ tests or one-way ANOVA, depending on whether data were normally distributed or not. $P$-values < 0.05 were considered statistically significant. Error bars represent SD of the mean. N, the number of independent experiments. An average of 15 cells were analyzed in each independent experiment. Statistical analysis was conducted on the averages from independent experiments.

## Step-by-step workflow of TWS protocol for segmenting mitochondrial cristae

Generate 16-bit Airyscan images of mitochondria showing cristae (Fig 12A, training image Supplemental Data 1). Next, open an Airyscan image and perform a background subtraction with a rolling ball filter = 50. Note: the scales of the images do not need to be manually specified, as this information is contained in the

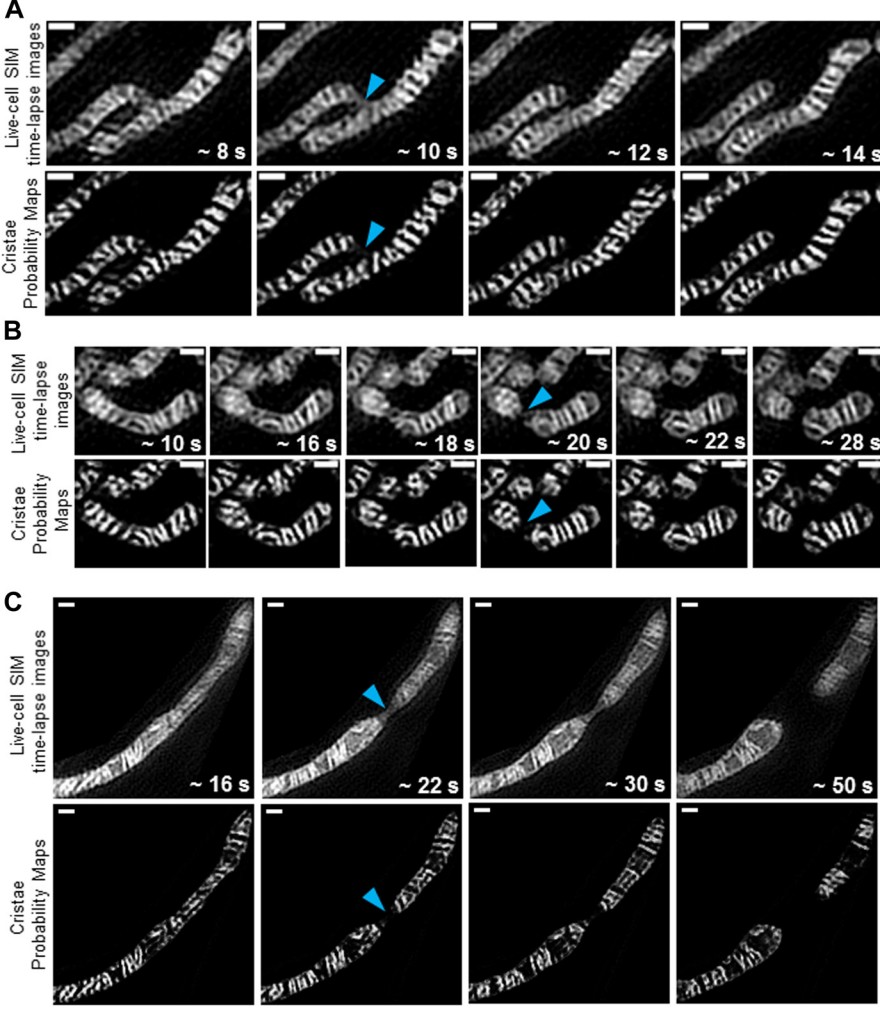

Figure 10. Trainable Weka Segmentation protocol shows fission sites containing decreased cristae density.
**(A, B, C)** Representative time-lapse SIM images of HeLa cells stained with MTG (top rows) with cristae probability maps (bottom rows), showing effective segmentation of cristae during mitochondrial fission events. Note the narrowing of the inner boundary membranes (blue arrowheads) before mitochondrial fission into two daughter mitochondria. Also note the decreased cristae density at fission sites. Scale bars = 500 nm. N = 3 independent experiments.

metadata of the image. Then, open the TWS plugin in the online Fiji software (Hall, 2009; Arganda-Carreras et al, 2017) by going to the (Fiji Is Just) ImageJ menu bar and clicking "Plugins" → "Segmentation" → "Trainable Weka Segmentation." After loading an Airyscan image of mitochondria into the plugin window, proceed with the following steps to train the cristae classifier:

1. On the left-hand side of the window, click "Settings" to open the Segmentation settings dialog box and select specific Training features (e.g., "Sobel filter," an edge-detecting algorithm, "Membrane projections," a feature that enhances membranous structures, and "Gaussian blur," which reduces noise) to determine the manner in which the machine-learning protocol segments objects within images.
2. In the Segmentation settings dialog box, create two classes of objects: Class 1 "Cristae"; Class 2 "Background." Click OK.
3. Use the cursor to mark structures that you identify as cristae (Fig S5A) and click on the "Add to Cristae" box on the right-hand side of the TWS window. Next, use the cursor to mark regions that you identify as background (i.e., not cristae) and click on the "Add to Background" box on the right-hand side of the TWS window.

4. Repeat this process numerous times on different regions identified by the user as cristae or as background within the same cell.
5. Click the "Train classifier" button on the left-hand side of the TWS window to initiate the training process. After the training is finished, an overlay image will appear, showing the two classes: "Cristae" as reddish brown and "Background" as green areas (Fig S5B). Note: this process can be repeated as many times as necessary until the user is satisfied with the accuracy of the segmentation. Click "Toggle overlay" to assess the progress of the training, as needed.
6. After finishing with training the classifier, click the "Save data" button on the left-hand side of the TWS window. Saving the data associated with TWS training will generate an .arff file. Next, click the "Save classifier" button on the left-hand side of the TWS window. This will generate a separate .model file. Note: to train the TWS classifier on additional images, it is necessary to load the previous .arff file into a new TWS window to build upon the data from previous training sessions.
7. Close the TWS window.

After training the classifier to effectively segment cristae from Airyscan images, proceed with the following steps to analyze the cristae architecture:

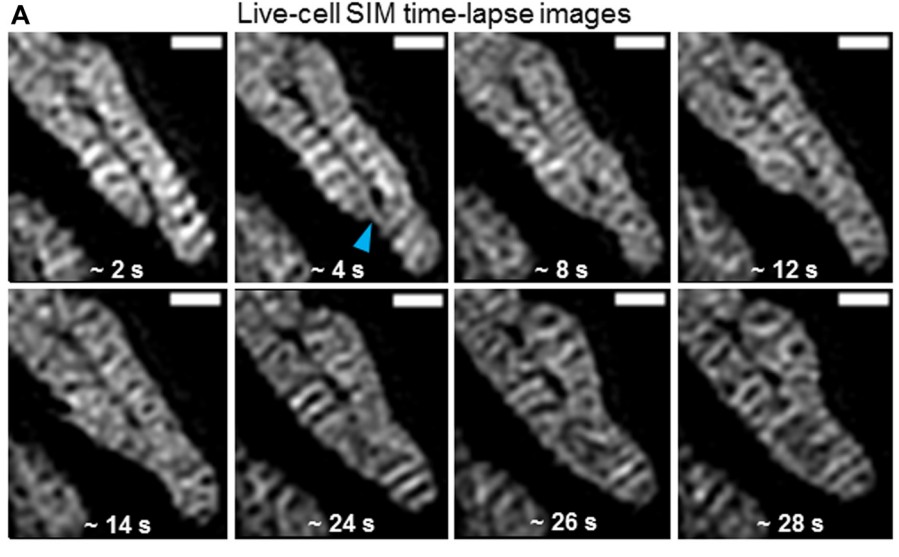

**A** Live-cell SIM time-lapse images

**Figure 11. Trainable Weka Segmentation protocol can quantify cristae remodeling during mitochondrial fusion events.**
**(A)** Representative time-lapse SIM images of HeLa cells stained with MTG. Scale bars = 500 nm. Note at ~4 s, a finger-like region of inner mitochondrial membrane extends from the tip of the mitochondrion on the left (blue arrowhead), before fusion with mitochondrion on the right in the following frame. **(B)** Cristae probability maps of time-lapse images from (A). Scale bars = 500 nm. Note at ~24 s, the separate cristae (red and green arrowheads) of the adjoining mitochondria have circularities of 0.7 and 0.5, respectively; however, after fusion in the following frame, the crista shows a circularity of 0.2, marking a transition from a less to more branching structure.

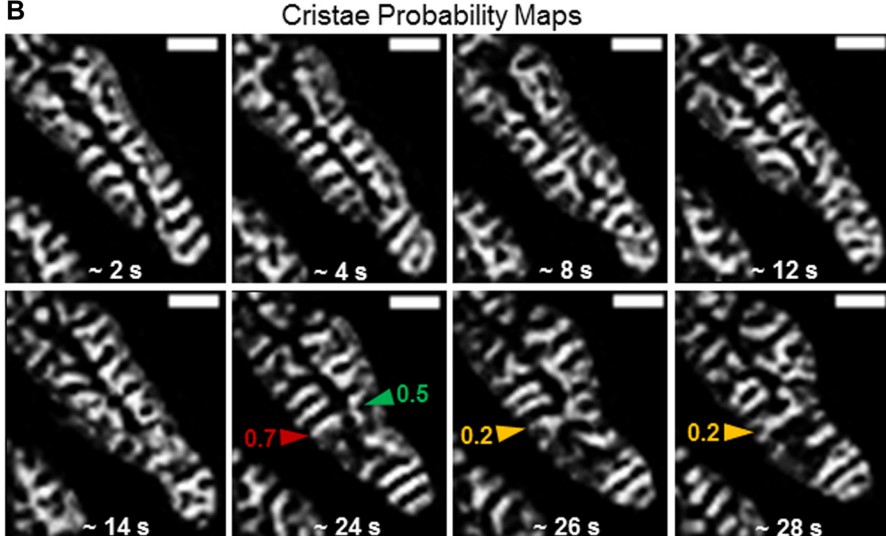

**B** Cristae Probability Maps

8. Open an Airyscan image (with background subtracted, as indicated above).

9. Duplicate this image (Ctrl + Shift + D) for a later stage of analysis.

10. In the (Fiji Is Just) ImageJ menu bar, click "Plugins" → "Segmentation" → "Trainable Weka Segmentation."

11. Click "Load classifier" on the left-hand side of the TWS window to load finished cristae classifier, created from training sessions in the previous section (steps 1–7).

12. Click "Get probability" on the left-hand side of the TWS window, which generates a stack of images showing the probability that each pixel belongs to a particular class of objects (Fig 12B). Note that, in channel 1, the white pixels correspond to areas that the classifier determined to be probable cristae structures, whereas the black regions belong to background. The pixels in channel 2 (not shown) are the inverse of those in channel 1.

13. Next, from the (Fiji Is Just) ImageJ menu bar, click "Image" → "Stacks" → "Stack to Images." Note: Because the channel 2 image

(i.e., the probability map of the background) is not required for further analysis, close it.

14. Click on the image containing the cristae probability map, and then, from the (Fiji Is Just) ImageJ menu bar, click "Image" → "Adjust" → "Threshold…"

15. From the Threshold window, adjust threshold until cristae are accurately distinguished from background (Fig 12C). Click apply.

16. In the Thresholder window, click "Convert to Mask."

17. In a new window, a binary mask will appear, showing cristae in black and background in white pixels (Fig 12D). In the (Fiji Is Just) ImageJ menu bar, click "Analyze" → "Analyze Particles…"

18. In the Analyze Particles dialog box, exclude particles that would be below the theoretical area of cristae, for example, 0.017 $\mu m^2$ and click OK. Note: determining an appropriate theoretical limit of cristae area will depend on cell type and/or the resolution of the microscope used to measure the ultrastructure.

19. A new window showing ROIs will appear, outlining cristae in yellow traces (Fig 12E).

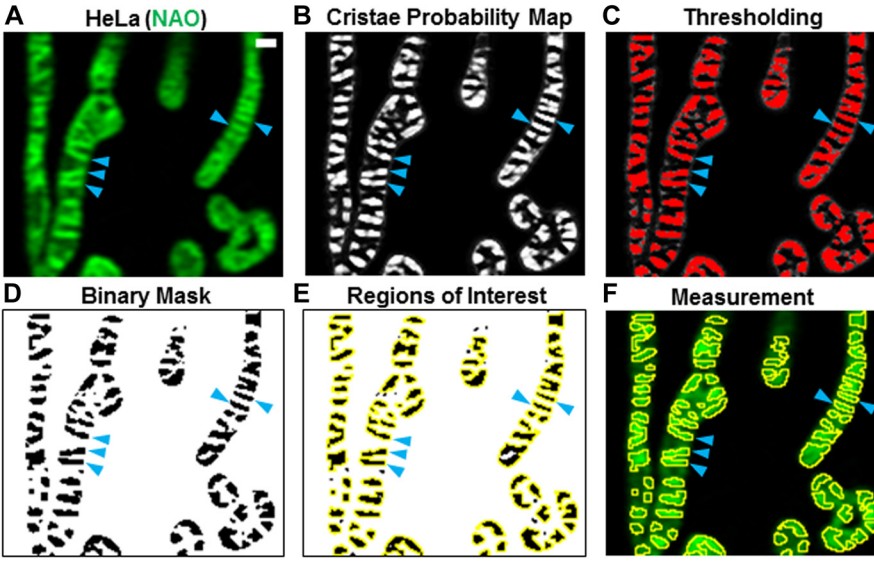

**Figure 12. Machine-learning approach using Trainable Weka Segmentation can segment mitochondrial cristae in living cells.**
Approach for quantifying live-cell imaging of cristae using the LSM 880 with Airyscan. **(A)** Image of cristae (blue arrowheads) in HeLa cells stained with 10-*N*-nonyl acridine orange (NAO). Scale bar = 500 nm. **(B)** Cristae probability map depicting areas likely to be cristae (white pixels) versus background (black pixels). Blue arrowheads denote cristae. **(C)** Thresholding of the cristae probability map in (B). Selected cristae are shown in red (denoted by blue arrowheads). **(D)** Binary mask resulting from application of thresholding step in (C). Blue arrowheads denote cristae (black pixels). **(E)** Regions of interest (ROIs) marked by yellow borders around cristae (black pixels). Blue arrowheads denote cristae. **(F)** Superimposition of ROIs onto original image in (A). Subsequent measurement yields data relating to various parameters, such as cristae density, area, and shape.

20. Next, click on the copy of the original image, made in step 9. Then, in the ROI Manager window, unclick and click the "Show All" checkbox. This will apply the ROIs generated from the binary mask to the cristae in the original image (Fig 12F).
21. Click "Measure" on the ROI Manager window.
22. A new Results window will appear, showing values corresponding to various parameters, such as Area, Mean, Perim. Circ., AR, etc. To change these readouts, go to "Set Measurements..." under the "Analyze" tab in the (Fiji Is Just) ImageJ menu bar. Note: Area, Circ., and other readouts are typically provided by default in the Set Measurements window. To make sure these measurements are provided, check the "Shape descriptors" box in the Set Measurements window. Copy and paste the data into Microsoft Excel. To obtain cristae density measurements, also segment the mitochondrial network and divide the total number of cristae by the total mitochondrial area (square micrometers) per cell. For each new image/cell, create a separate tab in Excel.

Open new images into the TWS window and apply the classifier to make additional measurements from different samples/assays. Overall, this workflow appeared to be effective at segmenting cristae from living cells. For analysis of SIM images of mitochondrial cristae, we followed this same basic TWS protocol. For technical assistance, including providing macros associated with this study, please contact primary authors.

# Supplementary Information

# Acknowledgements

We thank Drs Martin Picard, Barbara Corkey, David Nicholls, Gulcin Pekkurnaz, Gilad Twig, Ophry Pines, Victor Darley-Usmar, György Hajnóczky, Fernando Abdulkader, and Daniel Dagan for fruitful discussions. We also thank Drs Mingqi Han and Sean T Bailey for help with reagents and cell lines. OS Shirihai is funded by NIH-NIDDK 5-RO1DK099618-02. M Liesa is funded by UCLA Department of Medicine Chair commitment and UCSD/UCLA Diabetes Research Center grant, NIH P30 DK063491. DB Shackelford is funded by an NIH/NCI R01 CA208642-01. NW Hultgren and DS Williams are supported by R01 EY027442 and R01 EY013408 (to DS Williams). DS Williams also receives funding from P30 EY000331 (to DS Williams), and NW Hultgren is additionally supported by UCLA Postdocs Longitudinal Investment in Faculty Training (UPLIFT) K12GM106996 (to Michael F Carey).

## Author Contributions

M Segawa: conceptualization, data curation, software, formal analysis, validation, investigation, visualization, methodology, project administration, and writing—original draft, review, and editing.
DM Wolf: conceptualization, data curation, formal analysis, validation, investigation, visualization, methodology, project administration, and writing—original draft, review, and editing.
NW Hultgren: investigation.
DS Williams: resources.
AM van der Bliek: resources.
DB Shackelford: resources.
M Liesa: resources, supervision, and funding acquisition.
OS Shirihai: resources, supervision, funding acquisition, and writing—review and editing.

## Conflict of Interest Statement

The authors declare that they have no conflict of interest.

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
