## [Reviewer comments · Life Science Alliance]

Life Science Alliance

Quantifying cristae architecture reveals time-dependent characteristics of individual mitochondria

Mayuko Segawa, Dane Wolf, Nan Hultgren, David Williams, Alexander Van der Bliek, David Shackelford, Marc Liesa, and Orian Shirihai

DOI: <https://doi.org/10.26508/lsa.201900620>

Corresponding author(s): Orian Shirihai, University of California Los Angeles

Review Timeline:

Submission Date:	2019-12-02
Editorial Decision:	2019-12-23
Revision Received:	2020-04-27
Editorial Decision:	2020-05-12
Revision Received:	2020-05-14
Accepted:	2020-05-15

Scientific Editor: Andrea Leibfried

Transaction Report:

December 23, 2019

Re: Life Science Alliance manuscript #LSA-2019-00620-T

Prof. Orian Shirihai
University of California Los Angeles
650 Charles E. Young Dr. S
CHS 33-131
Los Angeles, CA 90095

Dear Dr. Shirihai,

Thank you for submitting your manuscript entitled "High-throughput quantification of mitochondrial cristae architecture and density in living cells" to Life Science Alliance. The manuscript was assessed by expert reviewers, whose comments are appended to this letter.

As you will see, your work received a mixed view from the reviewers. While reviewer #1 and #3 think that the method can get published here pending satisfactory revision, reviewer #2 thinks that the lack of benchmarking to existing methods makes it hard to appreciate the value provided. We think that you may be able to provide such benchmarking and would thus like to invite you to submit a revised version of your manuscript to us, addressing the reviewers' concerns. We would like to encourage you to also address the request of reviewer #2 to test your method for capturing dynamic changes (last point of the reviewer).

Thank you for this interesting contribution to Life Science Alliance. We are looking forward to receiving your revised manuscript.

Sincerely,

B. MANUSCRIPT ORGANIZATION AND FORMATTING:

Reviewer #1 (Comments to the Authors (Required)):

The report of Prof. Shirihai and colleagues describes the implementation of a semi-automated approach to quantify the architecture of the mitochondrial inner membrane.

The approach relies on the popular image processing package Fiji and several of its previously described plugins. Quantification of microscopic images is a challenging task and suitable software tools are urgently needed. Thereby this manuscript provides a valuable starting platform for researchers to analyze their own data sets. This protocol will certainly be valuable for the community.

I have only one serious concern that needs to be comprehensively addressed by the authors. It is claimed (page 5) that "the workflow was able to effectively segment cristae that were <50 nm apart from each other". This is a very bold and a very surprising statement, because the resolution of an AiryScan microscope is (at optimal conditions) around 120 nm. To my understanding it would be just impossible to resolve with such a microscope structures closer than 50 nm apart. This contradiction requires a detailed and careful explanation. The authors must provide several examples of a separation of structures closer than 50 nm and must provide the raw image data so that readers can verify this claim independently.

Minor concern

Related to the issue phrased above, it appears unlikely that the authors always recorded single cristae. It appears much more likely that often two or more adjacent cristae are perceived as a single crista. This should be mentioned.

Reviewer #2 (Comments to the Authors (Required)):

New methods for quantifying cristae in an unbiased way represent a new challenge in the field of mitochondrial research with considerable potential in the study and characterization of these organelles. Segawa et al reported a novel approach to efficiently characterize several features (size, shape) of mitochondrial cristae.

I have the concern that more than developing a new method, the authors simply show that an already existing method can be used to quantify cristae from high resolution microscopy images obtained with airyscan. Another important issue is how general is the claimed advantage of this protocol when applied to other high resolution microscopy techniques for living cells like STED or SIM.

Overall, their protocol efficiently selects and quantifies important aspects of mitochondrial cristae. However, in my opinion the advantage of this strategy with respect a conventional thresholding protocol is not clear. The authors argue that their method overcomes the limitations of conventional segmentation, but these limitations could also be overcome by a more restrictive thresholding algorithm. They should compare their approach with conventional segmentation under different algorithms of thresholding and demonstrate that their approach is still significantly more capable of detecting mitochondrial cristae shape size density etc.

Besides comparing their method to conventional thresholding, how do their results compare with cristae data obtained from EM on the same type of cells? They should also include this comparison for the cardiolipin deficient cells vs. wild type.

Moreover, the number of cells analysed in figures 5 and 7 should be increased in order to validate the approach for high -throughput analysis.

In addition, the analysis is done in living cells but with simple snaps, would be good to detect changes in cristae evolution over the time upon CCCP treatment or any alternative treatment.

Reviewer #3 (Comments to the Authors (Required)):

This is a very nice tool to analyze the latest *Cristae* fluorescence images semi-automatically after a training session.

However, some information and steps are still missing in the workflow. Is the requirement an 8bit image? Does the scale have to be entered beforehand? What to enter in Circularity?

To the authors: Please go through the workflow step by step again, maybe have someone with less experience run the macro in the lab. He/she will point to open questions.

If it is taken care of this, everything speaks for a quick publication.

Reviewer #1 (Comments to the Authors (Required)):

The report of Prof. Shirihai and colleagues describes the implementation of a semi-automated approach to quantify the architecture of the mitochondrial inner membrane.

The approach relies on the popular image processing package Fiji and several of its previously described plugins. Quantification of microscopic images is a challenging task and suitable software tools are urgently needed. Thereby this manuscript provides a valuable starting platform for researchers to analyze their own data sets. This protocol will certainly be valuable for the community.

We appreciate these encouraging comments and hope that our study will serve as an effective platform for quantifying the inner mitochondrial membrane.

I have only one serious concern that needs to be comprehensively addressed by the authors.

It is claimed (page 5) that "the workflow was able to effectively segment cristae that were <50 nm apart from each other". This is a very bold and a very surprising statement, because the resolution of an AiryScan microscope is (at optimal conditions) around 120 nm. To my understanding it would be just impossible to resolve with such a microscope structures closer than 50 nm apart.

This contradiction requires a detailed and careful explanation. The authors must provide several examples of a separation of structures closer than 50 nm and must provide the raw image data so that readers can verify this claim independently.

Thank you for bringing this to our attention! This has been corrected in the revised version. We see that our choice of words was poor in relation to this panel. We should not have implied that we were resolving cristae that were 50 nm apart, because this would be beyond the resolution of the Airyscan. For this panel, we simply wanted to note that our segmentation protocol could distinguish between objects, which have edges that are very close together. In other words, we did not want to claim that we were resolving two cristae that were in reality only 50 nm apart, since, to do this, we would have needed to measure the distance between the two peaks of the objects, in terms of their maximal fluorescence intensities. To address this point, we have removed this image and associated scale bar from the manuscript, because it was misleading. Furthermore, in the revised version, we refrained from deducing a resolution based on the images. Please also see the response below associated with the minor concern, because it is related to this point.

Minor concern

Related to the issue phrased above, it appears unlikely that the authors always recorded single cristae. It appears much more likely that often two or more adjacent cristae are perceived as a single crista. This should be mentioned.

We appreciate this criticism, which we have addressed in the revised manuscript. We further elaborated on this point on pgs. 14 and 15 in the Discussion: "It is important to note that, while state-of-the-art high- and superresolution live-cell imaging technologies are providing exciting new windows onto the complex biology of mitochondrial membranes, they remain, to date, unable to supersede EM as a way to probe the spatial dimensions of the organelle. For example, the resolution of electron

tomography (ET) systems typically ranges from 5 to 20 nm. Therefore, given average ET measurements of crista length and width to be 240 nm and 20 nm, respectively, the cross-sectional area of a crista is $4,800 \text{ nm}^2$ (or $0.0048 \text{ }\mu\text{m}^2$) (Mannella et al., 2013). The lateral resolution of SIM (including the GE DeltaVision OMX SR system used in this study) is approximately 100 nm. Our measurements of mean crista area taken from SIM micrographs were $0.0489 \text{ }\mu\text{m}^2$. This value is about one order of magnitude larger than estimates obtained from ET micrographs, a disparity consistent with an approximately 10-fold difference in the resolution of the two imaging technologies. Therefore, researchers should bear in mind that measurements of cristae area from live-cell, superresolution images are likely to be overestimated by roughly a factor of 10. Given that the mean crista-to-crista distance in HeLa cells is 120 nm (Stephan et al., 2019), it is likely that our Airyscan and SIM images did not fully capture all of the cristae within a particular frame – especially in cases where cristae were positioned more closely together than could be resolved by the imaging systems employed in this study.”

Reviewer #2 (Comments to the Authors (Required)):

New methods for quantifying cristae in an unbiased way represent a new challenge in the field of mitochondrial research with considerable potential in the study and characterization of these organelles. Segawa et al reported a novel approach to efficiently characterize several features (size, shape) of mitochondrial cristae.

I have the concern that more than developing a new method, the authors simply show that an already existing method can be used to quantify cristae from high resolution microscopy images obtained with airyscan. Another important issue is how general is the claimed advantage of this protocol when applied to other high resolution microscopy techniques for living cells like STED or SIM.

We thank the referee for bringing up this important point. To address whether our machine-learning segmentation of mitochondrial cristae would also be useful for superresolution images obtained with other microscopy techniques, we include new SIM images in the revised version. Using these images, we present a validation that our machine-learning protocol is still significantly better than conventional (manual) thresholding, when applied to SIM images of mitochondria. Please see the new Fig 6 and new Fig S3.

Overall, their protocol efficiently selects and quantifies important aspects of mitochondrial cristae. However, in my opinion the advantage of this strategy with respect a conventional thresholding protocol is not clear. The authors argue that their method overcomes the limitations of conventional segmentation, but these limitations could also be overcome by a more restrictive thresholding algorithm. They should compare their approach with conventional segmentation under different algorithms of thresholding and demonstrate that their approach is still significantly more capable of detecting mitochondrial cristae shape size density etc.

To address this point in the revised manuscript, we provide new data, where we examined more restrictive thresholding algorithms available in ImageJ (i.e., MaxEntropy and Shanbhag) in addition to our conventional (manual) thresholding. We determined that MaxEntropy performed about as well as conventional (manual) thresholding, and Shanbhag tended to underestimate the cristae density, and

tended to select only smaller regions within cristae area. Please see the new Figs 2-3 for details about the performance of these other benchmarks. Altogether, our machine-learning (TWS) approach provided better segmentation than any other algorithms that we tested.

Besides comparing their method to conventional thresholding, how do their results compare with cristae data obtained from EM on the same type of cells?

We thank the reviewer for this important comment. In the revised version of this manuscript, we addressed how our measurements of superresolution images of cristae compare to those performed on electron micrographs in the Discussion. Please see the text associated with the response to reviewer #1.

They should also include this comparison for the cardiolipin deficient cells vs. wild type.

We thank the reviewer for bringing up this point. We provide a comparison to EM images of PTPMT1-deficient cells on pg. 9 of the results section, where we note that EM images show swollen cristae, resembling what we observed using Airyscan microscopy. Numerical comparison to images from other studies was not feasible, since previous studies did not perform any quantifications of architectural parameters, and the images provided in these publications are insufficient for us to perform a thorough comparative analysis.

Moreover, the number of cells analysed in figures 5 and 7 should be increased in order to validate the approach for high-throughput analysis.

We thank the reviewer for requesting clarification on this point. In the graphs, each point represents an independent experiment. In each experiment, we analyzed approximately 15 different cells, including hundreds of mitochondria, and many thousands of cristae.

In addition, the analysis is done in living cells but with simple snaps, would be good to detect changes in cristae evolution over the time upon CCCP treatment or any alternative treatment.

We thank the referee for this making this wonderful suggestion as to detecting cristae dynamics over time, which led us to include the following exciting new data. In the revised manuscript, we addressed this point in two ways: 1) we performed a series of SIM experiments using FCCP on HeLa cells, where we tracked acute changes in cristae parameters over a period of 1 hr (please see new Fig. 8 and new Fig S4); and, 2) we examined changes in cristae architecture at high temporal resolution using SIM, examining changes in cristae structure independently of mitochondrial fusion and fission as well as in the context of classical mitochondrial dynamics (please see new Figs 9-12).

Reviewer #3 (Comments to the Authors (Required)):

This is a very nice tool to analyze the latest Cristae fluorescence images semi-automatically after a training session.

We thank the referee for these generous comments!

However, some information and steps are still missing in the workflow. Is the requirement an 8bit image?

We thank the referee for bringing up this point. In our study, we analyzed Airyscan images, which have a bit depth of 16, and SIM images, which have a bit depth of 15. It is likely that using images with higher bit depths will improve the ability to segment cristae. We added this information to the revised manuscript, in the new Supplementary Methods section.

Does the scale have to be entered beforehand?

To address this point, we modified the Supplementary Methods in the revised manuscript. To train the Trainable Weka Segmentation plugin to segment objects (e.g., cristae), it is not necessary to first enter the scale of the image. This information is retained in the metadata of each image and will be automatically applied during the machine-learning training process.

What to enter in Circularity?

Thank you for bringing up this point. Your point illustrated for us that we should clarify the specific steps required to make use of the software. We addressed this point in the revised manuscript, by addition of text to the Supplementary Methods section. Circularity is a shape descriptor that ImageJ provides as a standard readout or measurement of regions of interest (ROIs). To obtain values of circularity associated with specific ROIs, it is only necessary to make sure the "Shape descriptors" box is checked in the "Set Measurements" window of the "Analyze" tab. Typically, this box is checked by default.

To the authors: Please go through the workflow step by step again, maybe have someone with less experience run the macro in the lab. He/she will point to open questions.

We thank the referee for this suggestion, which we have gladly followed. We added additional explanations to the workflow (Supplementary Methods: pgs 29-32), following various recommendations of new users.

If it is taken care of this, everything speaks for a quick publication.

Thank you very much!

May 12, 2020

RE: Life Science Alliance Manuscript #LSA-2019-00620-TR

Prof. Orian S Shirihai
University of California Los Angeles
650 Charles E. Young Dr. South
CHS 27-200
Los Angeles, CA 90095

Dear Dr. Shirihai,

Thank you for submitting your revised manuscript entitled "Quantifying cristae architecture reveals time-dependent characteristics of individual mitochondria". As you will see, the reviewers appreciate the introduced changes and now support publication, pending minor revision. We would be happy to publish your paper in Life Science Alliance pending addressing the following:

- Please address the remaining reviewer concerns
- Please check one more time whether the author order in our system matches the one in the manuscript file
- Please incorporate the "Detailed, Step-by-step Workflow of TWS of mitochondrial cristae" in the main manuscript, Methods section, in order to make it easier for readers to apply your method

A. FINAL FILES:

-- Summary blurb (enter in submission system): A short text summarizing in a single sentence the study (max. 200 characters including spaces). This text is used in conjunction with the titles of

papers, hence should be informative and complementary to the title. It should describe the context and significance of the findings for a general readership; it should be written in the present tense and refer to the work in the third person. Author names should not be mentioned.

B. MANUSCRIPT ORGANIZATION AND FORMATTING:

Thank you for your attention to these final processing requirements.

Sincerely,

Reviewer #1 (Comments to the Authors (Required)):

The authors have addressed my concerns appropriately.

I do recommend publication of this manuscript.

Still, I am somewhat disappointed that in the discussion section they address correctly and in full the fact that with SIM and AiryScan microscopy presumably not all cristae are resolved, but miss to mention this entirely in the results sections. For example, on page 11 they state that they measure 1,415 +/- 257 cristae per HeLa cell. As discussed later, this number is presumably an underestimation. For readers who do not read the discussion, a pointer to this would certainly benefit the manuscript.

I would suggest to make the textual change.

Reviewer #2 (Comments to the Authors (Required)):

The authors have addressed the reviewers concerns adequately.

Reviewer #3 (Comments to the Authors (Required)):

The provided data convincingly demonstrate the power of high resolution live imaging of cristae and quantification of cristae dynamics.

This is a comprehensive manuscript almost ready.

Only some minor points have to be addressed beforehand.

1. it is discussed that the number of cristae are underestimated by light microscopy image analysis compared to EM micrograph analysis. Too close cristae are not resolved in SIM and Airyscan. if, as quoted that the average cristae to cristae distance is 120 nm (Stephan. et al. 2019, derived from STED images) with a resolution of 100 nm (p. 15 manuscript), all cristae should be discernable. However, EM analysis showed that the mean cristae distance in Hela cells is 50 nm (Wilkens et al., 2013), which is below the resolution. Please shortly discuss.

2. add scale bars to all images

3. an original imaging file in the supplements would be helpful to exercise the macro

4.p 32 step #5: left-hand site instead of right-hand site

5.p33 steps #9-10, convert to masks generates two images for cristae and background class.

These probably have to be separated from the stack

6. specify "analyze particles", which parameters shall be displayed, information about the circularity has not been added(?), which boxes activated to see the "yellow" surrounding?

May 15, 2020

RE: Life Science Alliance Manuscript #LSA-2019-00620-TRR

Prof. Orian S Shirihai
University of California Los Angeles
650 Charles E. Young Dr. South
CHS 27-200
Los Angeles, CA 90095

Dear Dr. Shirihai,

Thank you for submitting your Methods entitled "Quantifying cristae architecture reveals time-dependent characteristics of individual mitochondria". I appreciate the introduced changes and it is a pleasure to let you know that your manuscript is now accepted for publication in Life Science Alliance. Congratulations on this interesting work.

DISTRIBUTION OF MATERIALS:

Again, congratulations on a very nice paper. I hope you found the review process to be constructive and are pleased with how the manuscript was handled editorially. We look forward to future exciting submissions from your lab.

Sincerely,
